# Corneal Regeneration Using Gene Therapy Approaches

**DOI:** 10.3390/cells12091280

**Published:** 2023-04-28

**Authors:** Subhradeep Sarkar, Priyalakshmi Panikker, Sharon D’Souza, Rohit Shetty, Rajiv R. Mohan, Arkasubhra Ghosh

**Affiliations:** 1GROW Research Laboratory, Narayana Nethralaya Foundation, Bangalore 560099, Karnataka, India; 2Manipal Academy of Higher Education, Manipal 576104, Karnataka, India; 3Department of Cornea and Refractive Surgery, Narayana Nethralaya, Bangalore 560010, Karnataka, India; 4Harry S. Truman Memorial Veterans’ Hospital, Columbia, MO 65201, USA; 5One-Health Vision Research Program, Departments of Veterinary Medicine and Surgery and Biomedical Sciences, College of Veterinary Medicine, University of Missouri, Columbia, MO 65211, USA; 6Mason Eye Institute, School of Medicine, University of Missouri, Columbia, MO 65211, USA

**Keywords:** corneal dystrophies, corneal neovascularization, gene therapy, regenerative medicine, viral vectors, adeno-associated virus

## Abstract

One of the most remarkable advancements in medical treatments of corneal diseases in recent decades has been corneal transplantation. However, corneal transplants, including lamellar strategies, have their own set of challenges, such as graft rejection, delayed graft failure, shortage of donor corneas, repeated treatments, and post-surgical complications. Corneal defects and diseases are one of the leading causes of blindness globally; therefore, there is a need for gene-based interventions that may mitigate some of these challenges and help reduce the burden of blindness. Corneas being immune-advantaged, uniquely avascular, and transparent is ideal for gene therapy approaches. Well-established corneal surgical techniques as well as their ease of accessibility for examination and manipulation makes corneas suitable for in vivo and ex vivo gene therapy. In this review, we focus on the most recent advances in the area of corneal regeneration using gene therapy and on the strategies involved in the development of such therapies. We also discuss the challenges and potential of gene therapy for the treatment of corneal diseases. Additionally, we discuss the translational aspects of gene therapy, including different types of vectors, particularly focusing on recombinant AAV that may help advance targeted therapeutics for corneal defects and diseases.

## 1. Introduction

Corneal diseases are one of the leading causes of visual impairment globally [1]. A delicate physiological and functional balance is responsible for the transparency and clarity of the cornea. Therefore, any disease, acquired or genetic, that compromises this state of homeostasis can eventually lead to vision loss. Inherited, acquired, and iatrogenic corneal diseases, including corneal dystrophies, neovascularization, corneal scarring, haze, dry eyes, keratoconus, and corneal injuries, affect normal vision.

Approval of a gene therapy for retinal disease [2] has paved the way for corneal gene therapy; this method continues to develop and rapidly advance, and has a high potential for human application. Corneal gene therapy initially emerged in 1994, when corneal tissues were transduced successfully using replication-deficient adenovirus to treat acquired inflammatory disease [3]. Gene therapy for retinal diseases has made much greater progress when compared to corneal diseases. However, advancements made in the last two decades in the field of corneal gene therapy have brought this form of therapy much closer to clinical trials and application in human patients. Inherited corneal diseases, such as epithelial, stromal, and endothelial dystrophies [4,5], are natural candidates for corneal gene therapy. Complex conditions of multifactorial etiology, such as keratoconus, also have a genetic component associated to it, making it suitable for gene therapy [6].

The anatomic location of corneal epithelium makes it particularly attractive for non-invasive treatment by direct topical instillation of the gene delivery system [7]. Surgical, mechanical, chemical, and electric methods can also be used to administer gene therapy in the cornea. Furthermore, estimating the effectiveness and safety of the corneal treatment is easier due to the rapid and non-invasive visual observation using standard ophthalmologic methods [8]. Since the cornea is avascular and the eye has an altered immunologic profile in the intraocular compartments that usually dampens immune responses, the chances of systemic or even local immune reactions is lower.

The most striking advancement in the treatment of corneal diseases has been corneal transplantation and tissue engineering [9,10]. One of the most transplanted tissues worldwide is the cornea. In penetrating keratoplasty, the entire cornea is replaced, whereas in lamellar keratoplasty, only the damaged layers are replaced with requisite layers of donor tissue [10,11,12]. Unfortunately, in a subset of subjects, corneal transplantation has been associated with poor outcomes due to graft rejection and graft failure. Furthermore, the WHO has reported a shortage of corneal donors, due to which 10–15% of patients are reported to remain untreated [1,13]. Additionally, it has been reported that almost 53% of the world’s population lacks access to corneal transplantation, due to which there is only one cornea available for every seventy needed [14]. Due to the shortage of donor tissue and the fear of transmissible disease [14,15,16], there has been a continuous evolution of approaches in reviving corneal function and vision. An advantage of the cornea is its stability ex-vivo; it can be maintained in artificial physiological environment for weeks. This is especially advantageous for experimental purposes as well as for the administration of treatments prior to corneal transplantation, aiding in reduced chance of graft rejection [17,18].

To reduce global blindness, the development of a tissue-targeted gene-based intervention based on our understanding of molecular mechanisms that can be targeted by gene therapy is of paramount importance. Many of the diseases could be potentially treated using gene therapy by supplying a functional gene or changing the expression levels of specific genes in affected cellular layers. However, the majority of the studies in the field have been focused on the modulation of acquired corneal medical conditions. Regulation of the corneal microenvironment could be achieved using corneal gene therapy in different disorders by using induction or knock down of respective proteins. To improve the efficacy and safety of the treatment, local corneal gene delivery using a variety of vector and delivery modalities can be used to achieve low and continuous concentrations of the proteins [19]. Therefore, delivery of various immune modulators and growth factors to the cornea, in turn affecting local immune responses, inflammation, and proliferation, can be achieved without any systemic side effects. Corneal gene therapy has therefore been applied to the treatment of fibrotic disorders and corneal haze post-refractive procedures. Topical gene delivery to the cornea has the potential to effectuate local protein expression at concentrations unachievable by systemic administration of the transgenes or recombinant proteins [20]. Corneal gene therapy approaches can therefore enhance survival of corneal grafts and improve corneal diseases that currently require a corneal transplantation, thus preventing the need for an allograft [21].

In this review, we discuss genes associated with diverse corneal diseases that could serve as potential candidates for corneal gene therapy and the complexities linked with them. We also explore the strategies, challenges, and potential of gene therapy for the treatment of corneal diseases. Further, we discuss the multiple aspects of gene therapy that can help to improve personalized therapeutics in the field of corneal diseases.

## 2. Genes of Corneal Diseases

One of the first requirements for a successful gene-based treatment approach for the cornea is to identify the genes that need to be augmented, replaced, or edited, or identification of the pathways to be targeted by the protein product of the selected transgene.

### 2.1. Genes and Genetics of Inherited Diseases

Corneal dystrophies: Corneal dystrophies represent a heterogenous group of inherited corneal diseases that affect corneal development and cellular function. Corneal dystrophies are classified based on the layer of cornea that they affect. The human cornea essentially consists of the epithelium, the stroma, and the endothelium layer. Bowman’s layer, an acellular structure with densely compacted collagen fibrils, separates the epithelium and the anterior stroma. The Descemet’s membrane is the basement membrane of the endothelial layer that separates the posterior part of the stroma from the endothelium. Corneal dystrophies are classified into four main categories: epithelial and subepithelial dystrophies; Bowman’s layer dystrophies; stromal dystrophies; and Descemet’s membrane and endothelial dystrophies (Figure 1). Each of these dystrophies exhibit distinct clinical features, variable inheritance, age of onset, and course of progression [4,5] (Table 1). Corneal dystrophies are inherited as autosomal dominant, autosomal recessive, or X-linked.

*TGFβI*-associated dystrophies have been attributed to various mutants in *TGFβI* gene, with positions 124 and 555 being the most common. Deposition of TGFβ-induced (TGFβ) protein aggregates in the stroma or Bowman layer have been associated with these dystrophies [22]. Several uncommon mutations are scattered across five of the gene’s 17 exons and are associated with divergent clinical presentations of the dystrophy, including Reis–Bucklers corneal dystrophy, Thiel–Behnke corneal dystrophy, lattice corneal dystrophy type 1, granular corneal dystrophy type 1, and granular corneal dystrophy type 2. Epithelial recurrent erosion dystrophy (ERED) is caused by mutation in the *COL17A1* gene [23]. Table 1 shows the current international classification of most corneal dystrophies. Furthermore, sequencing analyses showed that mutations in *COL17A1* were causative of ERED [23]. Until the discovery of this mutation, Thiel–Behnke corneal dystrophy was thought to result only from mutations in the *TGFBI* gene. Heterozygous mutations in the *KRT3* or *KRT12* genes, encoding corneal specific Keratins 3 and 12, respectively [24], have been associated with Meesmann epithelial corneal dystrophy (MECD), a rare autosomal dominant inherited disease. Malfunction of K3 and K12 has been shown to cause mechanical fragility of the anterior corneal epithelium [25]. Various studies have identified at least 24 mutations for MECD, most of which are missense point mutation [24,26,27,28]. Early onset Fuchs endothelial corneal dystrophy (FECD) has been associated with mutations in the *COL8A2*, *SLC4A11*, *ZEB1*, and *LOXHD1* genes. The majority of FECD cases are caused by a trinucleotide repeat expansion in the *TCF4* gene [29], leading to altered mRNA processing due to sequestration of splicing factor proteins (MBNL1 and MBNL2) to the nuclear RNA foci.

Other inherited diseases: Other inherited disorders affecting the cornea include Aniridia and Mucopolysaccharidosis (MPS). Aniridia exhibits a dominant autosomal inheritance pattern with variable expression in several members of a family. A diverse set of mutations leading to haploinsufficiency of the *PAX6* gene, which is expressed in various regions of the eye including the cornea [30], have been demonstrated to be associated this disorder. MPS is a group of inherited metabolic disorders caused by the absence or malfunctioning of lysosomal enzymes that are responsible for the degradation of glycosaminoglycans. Lysosomal accumulation of undegraded glycosaminoglycans in keratocytes causes corneal clouding. MPS demonstrate a high range of clinical manifestations and are classified based on the enzyme that is dysregulated. Although common to most types of MPS (I-IX), corneal manifestations are most common in MPS I, VI, and VII. MPS I is a monogenetic disease caused by loss of function mutations in both copies of the IDUA gene. MPS VII, or Sly syndrome, is caused by mutation in the enzyme β-glucuronidase [31]; MPS VI, or Maroteaux–Lamy syndrome, is caused by mutations leading to dysfunction of the enzyme aryl sulfatase B, causing corneal clouding [32].

### 2.2. Genes Associated with Acquired Corneal Conditions

#### 2.2.1. Corneal Wound Healing

Disparate injuries can cause fibrotic scarring. Irrespective of the cause of the injury, transforming growth factor β (TGFβ) has been associated with aberrant corneal healing responses. Previous studies have demonstrated the interaction of the TGF-β superfamily proteins with the Smad family to activate downstream signaling [33]. Bone morphogenic protein is a part of TGF-β superfamily that appears to modulate keratocyte proliferation, differentiation, and apoptosis, thereby playing an essential role during corneal wound healing [34]. Studies have shown the important role of BMP7 in the development of the mammalian kidney and eye [35]. The complex wound healing signaling network involves the binding of BMP7 to type I and II receptors, allowing regulation of receptor-regulated Smads (Smad 1, 5, and 8) and inhibitory Smads (Smad 6 and 7) [36]. Improved healing and lessened scarring by overexpression of the inhibitory protein Smad7, thereby causing inhibition of TGFβ signaling pathway, have been observed in animal models [37]. Additionally, hepatocyte growth factor (HGF) and its receptor (c-Met receptor tyrosine kinase) have also been shown to play important roles in normal and healing corneal epithelium and keratocytes in the anterior stroma in vivo [38]. Upregulation of HGF in keratocytes has been observed upon injury to the corneal epithelium [39].

#### 2.2.2. Corneal Neovascularization

Corneal neovascularization occurs in various corneal pathologies, including inflammatory diseases, congenital diseases, contact lens-related hypoxia, inflammatory diseases, autoimmune diseases, chemical burns, trauma, corneal graft rejection, and infectious keratitis [40], which may lead to significant visual impairment or blindness [41]. Various angiogenic factors, such as VEGF, basic fibroblast growth factor (bFGF), matrix metalloproteinase (MMP), platelet-derived growth factors (PDGFs), and interleukin-1 (IL-1), mediate corneal neovascularization. In the retina and the cornea, VEGF-A is the predominant VEGF member driving pathologic neovascularization and is therefore a potential target of several drugs. The VEGF-A binds to two members of receptor tyrosine kinase family, VEGF receptors VEGFR1 and VEGFR2, also known as Flt-1 and KDR, respectively. Vascular endothelial cells (VECs) significantly express both of these receptors, and their expression increases in the presence of inflammation. It has been found that their activation promotes vascular leakage, VEC liberation, and VEC proteolysis [42,43]. Targeting these factors and inhibiting their expression or increasing expression of anti-angiogenic factors may help inhibit angiogenesis. Bevacizumab and ranibizumab are currently used in clinical settings to inhibit VEGF-A signaling pathways [44]. When mixed with non-liposomal lipid and delivered by means of subconjunctival injection, the brain-specific angiogenesis inhibitor 1 (*BAI1-ECR*) gene showed an effective reduction of corneal neovascularization [45]. Several of these targets that mediate neovascularization, including *Flt23k*, *Flt-1*, *PEDF*, *VEGFR Flt-1*, *MMP-9*, and *vasohibin-1* [46,47,48,49,50,51], could serve as a potential target candidates for corneal gene therapy.

#### 2.2.3. Corneal Graft Survival

Immunological rejection has always been one of the primary causes for corneal graft rejection [52]. Several different groups have shown significant prolongation of corneal allograft survival using various different transgenes and vectors in a variety of animal models [53]. Several of these transgenes lower immune response [54,55,56,57,58] and reduce angiogenesis, thereby preventing likelihood of donor graft tissue rejection due to inflammation, corneal scarring, and edema [59]. The number of transgenes that have shown success in modulating corneal graft rejection indicates the complex nature of the process and the multiple pathways involved. Prevention of activation of T cells by gene transfer of Cytotoxic T-Lymphocyte Antigen 4 protein (CTLA4-Ig) has shown to effectively prolong graft survival [56,60]. Furthermore, knocking down neuropilin-2 through RNA interference (RNAi) [61] has been shown to reduce the amount of free vascular endothelial growth factor-A (VEGF-A) [62], leading to a decrease in activated T-cell influx and improvement in graft survival.

#### 2.2.4. Multifactorial and Polygenic Diseases

A variety of conditions are associated with reduced central corneal thickness (CCT) which include keratoconus, keratoglobus, brittle cornea syndrome, Ehlers–Danlos syndrome, osteogenesis imperfecta, and myopia [63]. CCT reduction is also associated with various genetic determinants in the context of these diseases, although various systemic and environmental factors are also involved in disease. The most common among these conditions is keratoconus (KC), a complex multifactorial disease associated with a wide variety of etiological factors such as genetic predisposition, environmental insults, oxidative stress, and mechanical injuries or eye rubbing. Alterations in several biochemical mediators have been associated to KC pathogenesis, including lysyl oxidase (LOX) [64,65], MMP2 [66], MMP9 [65,67], and various collagens, including I, III, IV, V, VI, and VII [68]. Familial genetic studies and genome-wide studies of the KC families have identified genomic heterogeneity in the genomic loci among these families [69,70,71,72,73]. Mutations in *DOCK9*, *FLG*, *TGFβI*, *SOD1*, *ZEB1*, and *VSX1* genes were also found to be responsible for KC prognosis in some selected populations around the world [74,75,76,77,78,79].

## 3. Clinical Presentation of Various Corneal Diseases

Corneal disease is a major public health problem primarily attributed to infections, trauma, corneal scars, corneal dystrophies, and degenerative and multifactorial conditions [80]. Corneal dystrophies are a group of inherited diseases that can involve various layers of the cornea and are usually bilateral and progressive, resulting in progressive loss of the transparency of the cornea and visual deterioration [81]. They affect around 0.09% of the global population, with a majority of cases being of endothelial origin [82]. These conditions usually present at the clinic with a progressive decrease in vision and can also come with additional problems, such as severe photophobia and irritation due to recurrent corneal erosions (RCE) seen in the more superficial corneal involvement. Slit lamp examination aids the clinical diagnosis by the classical features observed. Episodes of RCE can have a deleterious effect on the quality of life of these patients and can also result in visual deterioration due to subepithelial or stromal scarring.

The deposition of various substances in the cornea also results in early visual impairment requiring surgical intervention. Certain dystrophies, such as the congenital stromal dystrophy and congenital hereditary endothelial dystrophy (CHED), result in amblyopia in these patients [83]; in Fuch’s endothelial dystrophy, there is intrastromal fluid collection, causing corneal scarring in advanced disease [84,85].

One of the most common dystrophic corneal conditions is Keratoconus, an ectatic disease characterized by the thinning of the corneal stroma; the disease is usually bilateral and can be progressive in nature. It has a typical onset in the first to second decade of age and can progress until the third to fourth decade. As the disease progresses, there is a significant deterioration in vision due to the irregular astigmatism and ectasia [86]. In advanced stages, there can be varying degrees of corneal scarring and even rupture of the Descemet’s membrane, resulting in corneal hydrops [87].

Corneal scars can develop after infections, trauma, and certain surgeries, including refractive procedures such as photorefractive keratectomy (PRK), or collagen crosslinking for KC [88,89]. This results in suboptimal visual outcomes.

## 4. Clinical Treatment Options Currently Available

The treatment of corneal dystrophies depends on the degree of involvement of the cornea and stage of the disease. Surgical treatment options include phototherapeutic keratectomy (PTK) in more superficial involvement and lamellar or full thickness corneal transplantation in more advanced disease.

Various modalities of treatment are available for other corneal conditions, such as KC, depending on the stage of the disease. Collagen cross linking is the primary method of management for progressive KC and can be combined with adjunctive procedures such as topography guided ablation and intracorneal ring segments. In advanced KC with scarring, treatment options include specialty contact lenses such as the scleral lenses, which are effective but may not be tolerated by all patients, or a lamellar or full thickness corneal transplant depending on the depth and extent of scarring [90]. With a prevalence of 1.38 per 1000 [91], many patients are at risk of becoming permanently blind. In addition, since the patients are typically affected during their productive age of 15–40 years, the socio-economic burden of KC is very high [92]. Furthermore, patients with KC experience a significant impact on their quality of life. A higher probability of having a psychiatric disorder has been associated with more severe cases of KC.

Visual rehabilitation for corneal scars includes the use of scleral contact lenses, photorefractive or phototherapeutic keratectomy using an excimer laser, or a keratoplasty, depending on the depth and position of the scar [88,93]. When the scar involves the entire corneal thickness, it requires a penetrating keratoplasty, while scars or opacities limited to a section of the cornea can be managed by a lamellar keratoplasty of the anterior or posterior cornea as required [11].

## 5. Current Challenges in the Treatment of Corneal Diseases

The available surgical options are effective in treating these conditions but do pose certain clinical and logistical challenges. Phototherapeutic keratectomy (PTK) is a useful and effective technique using excimer laser ablation to treat superficial corneal opacities secondary to infections, trauma, or dystrophy [10]. However, there can be recurrence of the primary pathology post-procedure, especially in the case of corneal dystrophies. Other possible complications post-procedure include induced refractive errors, infections, and corneal scarring [94].

One of the most important methods for treating various corneal conditions is corneal transplantation. Lamellar techniques of corneal transplantation have markedly improved outcomes for corneal grafts [95]. However, despite the rising popularity of these transplantation techniques, there are still several challenges faced with corneal transplants. A major problem faced in eye banking and corneal transplantation is the availability of viable corneal tissue for transplantation [80]. There is large disparity between the tissue requirement and the supply of corneal tissues, with a more than 4-fold difference between available tissue for transplantation and actual required numbers [14,96]. There is also the rejection of potential donor tissues during the screening process due to transmissible diseases such HIV and Hepatitis B and C [15,16]. Additionally, it is estimated that more than 50% of the world’s population does not have access to the surgical expertise required for corneal transplantation [1].

Another challenge faced in corneal transplantation is the possibility of graft rejections and failure. Although the early graft survival of full thickness corneal transplant is more than 70% 1 year post surgery, this survival rate deteriorates to less than 50% at 5 years, even in primary grafts [97,98]. This is further reduced in repeat grafts and high-risk grafts. Lamellar techniques of corneal transplantation, such as DALK, DSEK, and DMEK, have less risk of rejection and better surgical and visual outcomes in low-risk grafts [99,100]. In high-risk grafts, there is loss of the cornea’s immune privilege due to various preexisting conditions; treatment may involve long term immunomodulation with attendant complications and vary despite it [99,101]. In developing nations, there is also the problem related to cost of the procedure, long term follow-ups, and the availability of surgical and technical expertise [102,103].

## 6. Gene Therapy for Corneal Diseases

Gene therapy is a technique in which replacing or inactivating (knocking out) the mutated gene occurs as a result of targeted therapeutic delivery of correct nucleic acid directly into the patient’s cells. Gene therapy to treat any inherited corneal dystrophies works via three approaches: (a) Gene inactivation or silencing of the mutated gene that manifests a toxic effect on the cells; (b) Gene correction or replacement of the mutated gene with a normal copy of a healthy gene; and (c) Addition of a healthy copy of the gene that will ectopically express the therapeutic protein to rescue the disease phenotype. Gene therapy has a growing potential in the field of corneal dystrophies due to three major characteristics: accessibility in terms of injections and surgical interventions; its partial immune-privileged properties that limit immune responses towards the antigenicity of the transgene and the viral vectors; and the presence of a tight blood-retinal barrier that can help to prevent unintentional spreading or contamination of the neighboring tissues as well as to the general circulation. In addition, direct corneal delivery, either topically or via injection, can be achieved only in the affected eye in the case of unilateral conditions. The advancement in imaging technologies, such as pentacam and adaptive optics, further facilitates quantitative and qualitative evaluation of corneal changes after gene therapy. However, the wide heterogeneity in disease-causing genes in hereditary corneal disease requires knowledge and identification of the specific causative gene in each patient in order to consider gene therapy as an intervention. However, many of the pathways affected in the tissues as part of their pathology can be common. Therefore, there is a need to design differential strategies according to the causative genes, mutation, and inheritance pattern for gene therapy of various hereditary corneal dystrophies. The success of gene therapy for the treatment of any disease depends on the efficiency by which the therapeutic transgene is delivered to the target cell type. There are currently two main approaches in terms of vectors for delivering genetic material: viral and non-viral vectors.

### 6.1. Viral Vectors

Significant progress in understanding the molecular mechanism of diverse corneal diseases has led to the development of gene therapies in animal models. Gene therapy in animal models has demonstrated restoration of the biomechanical stability of the cornea by regulating the key proteins that are deficient in various corneal dystrophies, thus establishing a scientific basis for application in human subjects. Most studies on gene therapy of diverse corneal dystrophies have used viral vectors, including adeno-associated virus (AAV), lentivirus, and adenovirus, due to their unique ability to transduce a wide tropism of living cells with minimal resistance. The mechanisms by which the viral vectors deliver the therapeutic gene and translate the therapeutic protein into the host cells is discussed in this section.

#### 6.1.1. Adeno-Associated Virus (AAV)

AAV is a small (25-nm), non-pathogenic, nonenveloped virus that packages a linear single-stranded DNA genome of ~4.7 kb containing two genes that produce 4 Rep and 3 Cap proteins for replication and capsid proteins, respectively (Figure 2). The Rep gene encodes for four different proteins: rep40, rep52, rep68, and rep78. Rep68 and rep78 play an essential role in viral genome integration, replication, and transcriptional regulation of AAV gene expression, whereas rep 40 and rep 52 proteins are involved in viral genome encapsulation. The cap gene encodes three viral capsid proteins, VP1 (90 kDa), VP2 (72 kDa), and VP3 (60 kDa) (Figure 2), which are arranged in a 1:1:10 ratio and form an icosahedral symmetry. AAV belongs to the family Parvoviridae and is placed in the genus Dependovirus [104], as AAV requires a helper virus, such as adenovirus, to establish a productive infection cycle. However, recombinant AAV vectors used for gene therapy do not carry any of its native Rep/Cap genes or any helper genes, underlining its safety profile for therapeutic applications. Additionally, these vectors can transduce and express in both mitotic and post-mitotic cells.

The ability of the various rAAV serotypes to transduce a wide variety of ocular structures is due to its ability to bind primary cell surface receptors such as heparin sulphate proteoglycan (HSPGs) [105]. HSPGs are expressed in most of the cell types and uses integrin αvβ5 or fibroblast growth factor receptor (FGFR) as coreceptors for its internalization and endocytosis [106,107]. Once endocytosed, the viral particles are released from the endosome at a low pH [108,109]. The released SS-DNA of AAV is converted to ds-DNA by either annealing to another complementary strand of AAV or via the host cell DNA replication machinery. Subsequently, the viral genome remains as an episome and, with the help of host cell machinery, undergoes transcription and translation to produce the deficient therapeutic protein in the target cells over extended durations (Figure 2). This non-integrative nature of AAV vectors is another key safety feature wherein the risk of insertional mutagenesis in the host is low. A limitation of AAV vectors are their low cargo carrying capacity (Table 2).

The ability of the various rAAV serotypes to transduce ocular structures at the anterior segment of the eye has been extensively documented using vectors encoding marker proteins; therefore, it has become evident that a combination of serotypes, route of administration, and the choice of regulatory elements such as promotors allows the selective tropism of desired cell types. Recent studies have also shown that the corneal stroma is an ideal target for AAV-mediated gene therapy where the quiescent stromal keratocytes can receive the vectors and express the therapeutic proteins over long durations. In the context of corneal gene therapy, the first reported gene therapy was mediated by the AAV2 serotype, which demonstrated successful transgene delivery into a rabbit cornea in vivo [110]. The natural occurrence of the AAV2 serotypes may cause its pre-exposure in humans. This exposure leads to the production of neutralizing antibodies against the serotype. Thus, when a therapeutic gene packaged into a AAV2 serotype is delivered to the patient there is a chance of eliciting a humoral immune response against it which in turn may diminish the efficacy and safety of viruses carrying the therapeutic gene. The AAV5 serotype has been found to be the most divergent serotype [111], with a several-fold more efficient transduction efficiency than AAV2 [112]. In vivo gene therapy for corneal scarring using topical administration of the AAV5 serotype in rabbit models has shown a significant decrease in corneal haze and fibrosis, without any reports of any immunogenic or toxic immune response against it [113,114] (Table 3). In the domain of AAV vectors, AAV8 [115] and AAV9 [116] were considered the most efficient in corneal keratocyte transduction. Therefore, a chimeric AAV capsid was generated where an AAV8 capsid scaffold was engrafted with the AAV9 galactose binding domain. On evaluating the potency of infection of the chimeric AAV8/9 capsid following an intrastromal injection of the vectors into the human donor eyeballs, efficient widespread transduction of the transgene was observed. The AAV8/9 chimeric capsids were found to have greater and more efficient transduction efficiency when compared to either parent serotype at similar vector doses [117]. Therefore, a direct gene augmentation strategy using AAV vectors offers a viable option to cure diseases for which no treatment options exist.

#### 6.1.2. Lentivirus (LV)

Lentiviruses are single-stranded enveloped RNA viruses and are members of the retroviridae family. A mature LV is 100 nm in diameter and has a cylindrical core structure. LV transduction takes place through association with specific cell surface receptors. After binding to the receptor, the viral membrane fuses with the host cell membrane [126] and injects the nucleoprotein complex into the cell. After its entry, the viral RNA is reverse-transcribed into ds-DNA by the reverse transcriptase enzyme associated with the nucleoprotein complex. The ds-DNA enters the nucleus through the nuclear pore complex and is integrated into the host genome with the help of the viral integrase [127] (Figure 3). The major merit of using LVs is its larger transgene carrying capacity, broad tropism, and its ability to transduce both dividing and non-dividing cells (Table 2). Numerous ex vivo gene therapy experiments using LVs for corneal graft rejection and corneal fibrosis (Table 4) in various animal models have shown successfully enhanced transgene expression in the corneal epithelium, endothelium, and keratocytes and thus rescuing the disease phenotype. The cons of using LVs as a vector for corneal gene therapy are that it possesses high immune cell infiltration post-infection, random integration potential [128,129], and a consequent risk of insertional mutagenesis/teratoma formation. Due to these crucial shortcomings, the use of LVs still requires immense evaluation and identification of preferential integration sites in the host genome before being applicable to humans for therapies.

#### 6.1.3. Adenovirus (Ad)

Adenoviruses are double-stranded enveloped DNA viruses and are members of the Adenoviridae family. More than 50 known human adenoviral serotypes are present in nature, of which serotypes 2 and 5 are the most widely used in gene therapy. Ad vectors can transduce both dividing and non-dividing cells. It can harbor a therapeutic gene up to 30Kb in size and deliver it to the target tissue. Their inability to integrate into the host genome reduces the chances of insertional mutagenesis. In adenoviral vector preparations, high titers of pure viruses can be easily obtained from a single viral prep, thus a small volume of vector injection into the cornea is enough for a high level of transduction. These characteristic features of Ads make them an attractive candidate for gene therapy studies (Table 2). The Ad vectors deliver the therapeutic gene in the host cell through receptor-mediated endocytosis [135], by binding to the cell surface coxsackie adenovirus receptor, αvβ3 integrin, and clathrin-coated pits. Once internalized, the viral genome containing the therapeutic gene is released into the cytoplasm which further enters the nucleus through the nuclear pore complex and remains distinct from the host genome and is expressed as episomes to produce the therapeutic protein (Figure 3). Successful ex vivo and in vivo gene therapy of the cornea was first investigated by adenoviral vectors in different animal models (Table 5). However, the shortcomings were the short-lived expression of the delivered transgene, primarily due to the rapid histone association and silencing of the Ad genome. This necessitates repeated vector injection, which showed more toxic responses to the cells than the original exposure [136]. The Ad capsid structures (the pentons and heptons) are potent targets for rapid immune recognition causing Ad mediated gene therapy to typically induce significant host immune response [137]. Also, most humans are pre-exposed to wild-type adenovirus and therefore the immune responses are also mounted against the viruses, thereby giving a pre-existing immunity to the cells [17]. This ultimately dampens the efficacy of the vectors intended for the therapy (Table 2).

### 6.2. Non-Viral Vectors

The delivery of the therapeutic gene into the cornea can also be achieved using a non-viral origin vector mediated gene therapy. This includes the use of liposomes, compacted nanoparticles, electroporation, particle gun bombardment, and many other methods. The advantage of non-viral vectors is low immunogenicity, ease of manipulating their chemical properties to suit DNA delivery, the possibility of large-scale production, and transferring large vectors without any immune reactions. However, non-viral vectors may present obstacles such as inefficient cellular membrane transport, intracellular/extracellular degradation, and lack of long-term gene expression [146].

#### 6.2.1. Electroporation

Electroporation is a technique that uses short and intense electric pulses to create pores or reversibly permeabilize the cell membrane in order to deliver the naked circular DNA carrying the therapeutic gene into the target cells (Figure 4). This physical method of non-viral mediated gene therapy has been widely used for successfully delivering plasmid DNA into the corneal endothelium and stromal keratocytes [147,148]. The optimal field strength of 100–200 V/cm was found to be suitable for the transfer of naked plasmid DNA without any corneal damage or inflammation. Various in vivo experiments on different animal models for stromal keratitis and corneal endothelium wound healing experiments showed a 1000-fold increase in gene uptake in the cornea in comparison to injection of DNA alone, leading to inhibition of the disease phenotype [20,149]. One of the main drawbacks when using electroporation is that the permeabilization as a result of the electric pulse becomes irreversible due to the heat generated during the process [150].

#### 6.2.2. Nanoparticles

Nanoparticles are ultrafine particles that range between 1 and 100 nm in size and are widely used in nanomedicine due to their (a) small size, (b) ability to deliver the transgene into the intracellular compartment of the cells, (c) high surface area to volume ratio, (d) capability to deliver a larger payload, and (e) negligible toxic damage to the cell membrane. The use of nanoparticles is highly suitable for the delivery of transgene for the treatment of eye-related diseases as they can permeabilize across various ocular barriers including the cornea, sclera, conjunctiva, and, in some cases, the blood-retinal barriers. Nanoparticles can harbor multiple cargo types, including DNA, peptides, antibodies, molecular sensors, and drugs, in the desired cellular layers of the eye. The feasibility of using nanoparticles in various in vivo experiments to rescue the disease phenotype in corneal scarring has been demonstrated by several groups (Table 6). Nanoparticles are mainly classified as metallic, polymeric, and hybrid nanoparticles. Polymeric nanoparticles that are usually prepared from albumin, chitosan, and polyethyleneimine (PEI) have been found to be more efficient in delivering the transgene into rodent corneas in vivo without any significant side effects [151,152,153]. In vivo delivery of transgene using hybrid nanoparticles in rabbit cornea to treat corneal fibrosis has demonstrated an efficient cargo of transgene in the target cells with significant inhibition of fibrosis and no visible toxic effects [36].

The direct transfer of the therapeutic gene to the corneal cells in vivo has been achieved using cationic lipids [154]. The positively charged cationic lipids bind to negatively charged DNA molecules to form a lipid-DNA complex that has a high affinity for the cell membrane. The lipid-DNA complex is then endocytosed into an endocytic vesicle followed by its trafficking and release from the endosomal compartment. The nuclear uptake of the DNA takes place through the nuclear pore complex, which then forms an episome to express the therapeutic protein using the host cell machinery [155] (Figure 4). The effectiveness and safety of liposome-mediated therapeutic gene delivery in various ex vivo and in vitro studies has been demonstrated in the last few years (Table 6).

**Table 6 cells-12-01280-t006:** Corneal in vivo and ex vivo gene therapy using non-viral vectors.

Non-Viral Mediated Corneal Gene Therapy									
Vector	Disease	Gene	Promoter	Dosage	Model	Species	Mode of Administration	Outcome	Reference
Electroporation	Stromal Keratitis	*IL-10*	CMV, UbC	1 µL Plasmid solution (5 µg Plasmid DNA in 10 mM Tris, pH 8.0; 1 mM EDTA; and 140 mM NaCl)	In Vivo	Female Balb/c mice weighing 16 to 24 g	Stromal injection of plasmid DNA, followed by gold-plated Genetrode electrodes were placed on the cornea on either side of the area injected. An ECM 830 square wave electroporator was used to deliver eight pulses of 10-msec duration at a field strength of 200 V/cm.	Gene expression driven by the CMV promoter remained high for three days, which started to reduce by 2-fold each day thereafter. Replacing the promoter with UbC surprisingly showed similar half-life gene expression. Also, an adverse effect was observed when using DNA nuclear-targeting sequences in vectors.	[148]
CRISPR/dCas9	Corneal Endothelial wound healing	*SOX2*	Not mentioned	0.1 nmol	In Vivo	6–8 weeks old Sprague–Dawley rats; a 12-h light/dark cycle at 25 °C for 7 days before initiating experiments.	Anterior chamber injection of plasmid DNA, followed by 7-mm Tweezertrodes were placed on each cornea, with the positive electrode on the plasmid-injected eye. The parameters were set at 140 V, 100 milliseconds length, 950 milliseconds interval, five pulses, and 100 V/cm^2^	SOX2 activation promoted the reduction of central corneal thickness and corneal opacity in comparison to the control group. Additionally, an increase in Cell viability, proliferation rate, and the number of cells in the S-phase was observed after SOX2 overexpression.	[147]
CRISPR-Cas9	Granular corneal dystrophy (GCD 2)	*TGFBI*	Not mentioned	CRISPR-Cas9 constructs (2.5 µg per well) and ssODN (1 µg per well)	In Vitro	Human, GCD 2 patient-derived corneal keratocytes	Transfection	Effective gene correction efficiency of R124H mutation associated with GCD2 disorder was observed without any off-target effects. In heterozygous cells, the correction efficiency was found to be 20.6% and in homozygous 41.3% respectively.	[156]
Lipofection	Meesmann’s Epithelial Corneal Dystrophy (MECD)	*KRT12*	Not mentioned	200 ng plasmid (1 well of 12 well plate)	In Vitro	Corneal limbal epithelial cell derived from limbal biopsy of MECD patients	Lipofectamine 2000	Potent and specific knockdown of K12-Leu132Pro at both the mRNA and protein levels. An allele-specific knockdown of 63% of the endogenous mutant allele was observed.	[157]
Lipofection	lattice corneal dystrophy type I (LCDI)	*TGFBI-Arg124Cys*	CMV	200 ng plasmid (1 well of 12 well plate)	Ex Vivo	Corneal limbal epithelial cell derived from limbal biopsy of LCD1 patients	Lipofectamine 2000	The siRNA specific to TGFBI-Arg124Cys, efficiently suppressed the mutant allele	[158]
PEI gold nanoparticle	Corneal Fibrosis	*BMP7*	CAG	37.5 µL of 150 mM PEI2-GNPs with 10 µg of plasmid DNA	In Vivo	Female New Zealand White rabbit weighing 2.0 to 3.0 kg	Topical	Significant inhibition of fibrosis post-PRK was observed.	[36]
PEI nanoparticle	Corneal Fibrosis	*Decorin*	CAG	150 mM linear 22 kDa PEI+ 2 µg of plasmid	In Vitro	Horse	Topical	22 KDa PEI nanoparticle effectively inhibited TGFb-mediated fibrosis	[159]

### 6.3. Gene Therapy Strategies for Precise and Targeted Therapeutics

#### 6.3.1. Gene Augmentation

Gene augmentation therapy is a straightforward approach in which the transfer of the functional therapeutic gene into the affected cells takes place to restore the expression of an inadequately functioning gene [160,161] (Figure 5). This approach is mainly applicable to recessive genetic diseases. For the augmentation strategy to be effective, the augmented transgene must synthesize physiological and adequate levels of the normal protein and the tissue status and disease stage at which treatment is attempted should not be terminal. To minimize off-target transgene expression that may have a pathogenic effect within the cells, it is important to focus on the appropriate regulation of the transgene expression for augmentation-based approaches. To address appropriate transgene regulation, cell or tissue-specific promoter sequences can be designed to modulate the expression of the therapeutic transgene. For example, in response to inflammation, NFκB responsive promoter sequences have been used to drive the transgene expression in immune organs [162]. A commonly used approach in gene augmentation therapy for secretory protein is to target the delivery of the therapeutic gene to a distant tissue/cell and not the affected cell type or tissue; this will help in the production of the same deficient therapeutic secretory protein to rescue the disease phenotype. This approach can be adopted when (i) transduction of affected cells with the therapeutic gene may not be able to produce a physiologically relevant amount of secreted transgene protein; (ii) the affected site of gene delivery has been rendered sensitive to any further damage that can be caused during the gene delivery procedure; and (iii) the site of gene delivery is highly exposed to the host immune system which can lead to a transgene-directed immune response. Such an approach in corneal gene therapy has not yet been listed in the literature.

Gene augmentation therapy for corneal diseases has been conducted in various animal models in vivo to rescue the disease phenotype. There are numerous studies that have shown that gene augmentation using viral and non-viral vectors can help to significantly improve the disease pathology in murine, ovine, and canine models.

#### 6.3.2. Gene Editing

In recent times, gene editing has become one of the most utilized approaches in the field of gene therapy. This approach is able to target both autosomal dominant and recessively inherited corneal dystrophies that lead to loss of function due to mutation of the target gene. The strategy aims to restore/correct the normal functioning of the mutated gene by adding, removing, or altering the genome at the site of the mutation. One of the most specific and advanced genome editing technologies to date that has been used for gene correction in the anterior segment of the eye is Clustered Regularly Interspaced Short Palindromic Repeats (CRISPR) and CRISPR-associated (Cas) proteins, termed the CRISPR-Cas system [163]. Meganucleases, Zinc finger nucleases (ZFNs) [164], and transcription activator-like effector nucleases (TALENs) are also used in this approach. CRISPR-Cas-based genome editing has been found to be more favorable than the other mentioned nucleases due to its relatively simple design for targeting the mutated gene [165]. On the other hand, significant modifications in the DNA-binding protein domains are required to target with ZFNs and TALENs.

The mechanism of CRISPR-Cas 9 involves the binding of Cas9 nuclease at the genomic locus that is complementary to the guide RNA; this guide RNA creates a double-stranded break, which is repaired via two mechanisms: (i) error-prone non-homologous end joining (NHEJ) or (ii) homologous directed repair (HDR) (Figure 5). NHEJ can create insertions or deletions in the mutated gene, resulting in the formation of a premature stop codon. This can be used to knock out the expression of the truncated protein. HDR can help incorporate specific alterations into the mutated gene provided by the repair template. All studied Cas9 enzymes require a protospacer adjacent motif (PAM) next to the target site for target DNA recognition. Some recent studies have also shown the ability of certain divergent CRISPR-Cas9 enzymes that can recognize and cleave single-stranded DNA (ssDNA) using an RNA-guided, PAM-independent recognition mechanism [166]. Intravenous delivery of the CRISPR-Cas9 system cannot be used in the eye due to the blood-ocular barrier [167]. AAV as a vector has been widely used for the delivery of CRISPR-Cas9 construct into the target tissue/cell. However, the biggest shortcoming of using AAV vectors is their carrying capacity. They are often too small to accommodate the full genome editing system. To address this issue, a dual-AAV vector design has been implemented in which the Cas9 nuclease and the single-guide RNA (sgRNA) cassettes have been packaged in two separate vectors and delivered to the target tissue, which showed highly efficient genome editing in the hepatocyte cells [168]. The CRISPR-Cas 9 system has been used to treat corneal dystrophy in animal models. In one study, selective disruption of the mutated copy of the *KRT12* gene in a humanized MECD mouse model was achieved by targeting the protospacer adjacent motif (PAM) sequence that was generated by the missense mutation in the *KRT12* gene [169].

#### 6.3.3. Gene Silencing

Gene augmentation therapy is likely to be ineffective in rescuing the disease phenotype in autosomal dominant diseases. For augmentation therapy to work effectively, the suppression of the mutated gene expression must be addressed first. Silencing the mutated gene can be mediated via delivering a small double-stranded non-coding RNA construct that is designed to act via RNA interference (RNAi) [170]. RNAi promotes post-transcriptional gene silencing by enzymatic degradation of the complementary RNA species. This silencing is mediated by a large multi-component RNA/protein complex called the RNA-induced silencing complex (RISC) [171] (Figure 5). siRNAs (small interfering RNA) have been tested in various corneal dystrophies, including MECD, wound healing, and neovascularization [157,172,173].

In the context of mRNA-based gene therapy, the most successful approach is the use of antisense oligonucleotides (AONs). AONs are short single-stranded DNA or RNA that interacts with the complementary mRNA to block the translation by altering the pre-mRNA splicing [174]. The first AON approved by the FDA was for the treatment of cytomegalovirus retinitis: fomivirsen (Vitravene). In the cornea, the first phase clinical trial was conducted to explore the efficacy of a topically administered AON (Aganirsen) that targets the insulin substrate-1 receptor to block corneal neovascularization in keratitis patients. The AON administration was found effective and thus reduced the need for corneal transplantation [175]. GS-101 AON was also found to be a potent anti-angiogenic compound that helped in the significant regression of corneal neovascularization [176].

Despite gaining success, the strategy of gene silencing fails to completely silence the target gene expression. For the RNAi pathway, the major drawbacks, such as off-the-mark effects and extended toxicity of the small RNA molecules, must be considered and studied carefully for its successful application in clinical settings.

#### 6.3.4. Dual Vectors

Limitations of using rAAV-based gene therapy correctional strategies include its limited packaging size for genes larger than 4.7 kb such as PD-L1, which is widely used in corneal graft rejection studies. Several strategies have been designed to take advantage of the head-tail concatamerization of the AAV genome (Figure 6). One strategy is to use the overlapping sequence approach. In this method, efficient reconstitution and transgene expression rely on the use of two separate (dual) AAV vectors, each one carrying half of a large gene. Upon coinfection of the target cell from both vectors, the two halves will be reconstituted due to the canonical ability of AAV genomes to concatamerize via intermolecular recombination. In 2007, Ghosh et al. [177] developed a hybrid dual vector strategy to expand the packaging capacity in the rAAV system, the trans-splicing vectors, and showed effective, whole-body transduction using a mouse model of Duchenne Muscular Dystrophy. This trans-splicing strategy employs the use of a splice donor at the 3′ end of one-half of the target gene in one vector and a splice acceptor at the 5′ end of the other half of the transgene. This allows for efficient reconstitution of the mRNA transcripts by the head-to-tail concatamerization process. The third dual AAV approach (hybrid) is a combination of the two previous approaches; it is based on the addition of a highly recombinogenic exogenous sequence (recombinogenic region) to the trans-splicing vectors in order to increase their recombination efficiency. In 2011, Ghosh et al. [178] reported the use of minimized bridging sequences from the highly recombinogenic alkaline phosphatase (~227–247 bp) to circumvent the available space for packaging the transgene. In this approach, the authors demonstrated the complete reconstitution of the B-Galactosidase gene (LacZ) when packaged separately and co-infected to check expression. Expression of the reconstituted LacZ was proven to be better than the original hybrid strategy developed in suitable cell cultures and animal models.

### 6.4. Current Scenario of Corneal Gene Therapy

In the last two decades, research in the field of ocular gene therapy has advanced from a conceptual validation to a clinical reality with the approval of the first vision restoring gene therapy treatment for LCA (Leber congenital amaurosis) [2]. However, advances made in research related to the cornea have been considerably less when compared to the retina [179]. Progress in the field of corneal gene therapy has been steadily progressing despite remaining at the preclinical level. Several studies have identified various therapeutic genes and used several animal models to demonstrate the use of gene therapy in treating corneal defects [180,181].

Most of the preclinical work has focused mainly on treating and preventing acquired conditions such as corneal haze, corneal wound healing, herpes simplex keratitis (HSK), corneal neovascularization, and corneal graft rejection (Table 3, Table 4, Table 5 and Table 6). Studies have shown that moderate corneal conditions, such as thick haze and inflammation produced by keratectomy surgery with excimer laser effectively, can be treated by gene therapies in rabbits in vivo [114,182]. *Inhibitor of differentiation 3 (Id3)* gene, a transcriptional repressor, is known to inhibit differentiation of corneal keratocyte to myofibroblasts. Recently, a study showed the therapeutic effects of AAV5.Id3 gene therapy on corneal pathology and ocular health in rabbit eyes with corneal scarring/fibrosis induced by alkali trauma [125]. Further, dual gene therapy with *BMP7* and *HGF* effectively treated both fibrosis and neovascularization in a severe corneal opacity model produced by chemical injury [182]. Neovascularization in the host cornea is one of the primary reasons for corneal transplant rejection and therefore several studies have focused on reducing or preventing corneal neovascularization. TGFβ and VEGF growth factors play an important role in promoting fibrosis and neovascularization in the cornea in vivo. Decorin, a small leucine-rich proteoglycan, is a potent inhibitor of both these growth factors. AAV5 mediated decorin gene therapy has shown to effectively ameliorate corneal neovascularization and fibrosis in rabbit eyes in vivo [159,181,183]. Additionally, AAV5-Smad7 gene therapy has shown to inhibit corneal fibrosis post-PRK in rabbit stroma in vivo [113]. One study used a recombinant adeno-associated viral (rAAV) vector carrying an endostatin gene as an anti-angiogenic strategy to show successful inhibition of neovascularization when administrated by subconjunctival injection. Here, a minimal immune response with stable transgene expression was reported for 8 months [184]. Another study showed an effective reduction of corneal neovascularization when the *brain-specific angiogenesis inhibitor 1* (*BAI1-ECR*) gene, mixed with non-liposomal lipid, was delivered by means of subconjunctival injection in an in vivo rabbit model [45].

The time between donor cornea collection and recipient corneal transplant would be a perfect time to apply a pre-treatment to the cornea to reduce the chance of corneal graft rejection. Recently, rabbit and human donor corneal buttons were incubated ex vivo with an AAV vector-mediated *human leukocyte antigen G* (*HLA-G*). The treated group did not exhibit edema and neovascularization for more than 2.5 months upon allotransplantation. Furthermore, xenotransplantation (human donor to rabbit recipient) showed a delayed rejection time from 18 days to 29 days [121]. Delivery of IL-10 and IL-12 with adenoviral vectors to ovine corneas exhibited higher rates of graft survival [55,144,145,185]. Increased endothelial cell survival was observed with lentiviral vector-mediated gene delivery of baculoviral p35 or mammalian Bcl-xL at various stages of storage [186]. With continued investigations, gene therapy has the potential to improve survival of corneal grafts as well as overcoming the need for corneal transplantation.

HSK is considered a major determinant of corneal graft rejection after transplantation. Most approaches have focused on preventing the development of herpetic lesions and diseases by downregulating the viral gene expression and affecting viral replication. More recently, LAT-targeting ribozymes were delivered using AAV to block viral reactivation in the eyes of rabbits with latent HSV-1 infection. Viral activation was blocked in more than 60% of the infected eyes [187]. Another study employed a different approach to target latent HSV-1 by using rare-cutting endonucleases such as meganucleases, which can be engineered for increased specificity. They delivered anti-HSV 1 meganucleases using rAAV to human cornea ex vivo before transplantation to reduce the chance of reinfection and graft rejection [188].

Gene therapy for treating patients with MPS has been under investigation for the last three decades. While phase I/II gene therapy trials are ongoing for some types of MPS (MPS I, II, IIA, IIIB, and VI) [189], the influence of these therapies on corneal opacity with MPS remains unclear. There are several gene therapy-based strategies targeting the ocular manifestation of MPS, including both systemic and local approaches, but these investigations are limited to animal models. Potential prevention and reversal of corneal blindness was shown using AAV8G9-IDUA (an AAV8 capsid scaffold with AAV9 putative galactose binding domain) gene therapy [117,120,190]. Additionally, reduced corneal clouding was seen with adenoviral-mediated expression of *β-glucuronidase* (*GUSB*) in the stromal region in MPS VII mouse models [191]. Similar results were obtained following adenovirus-mediated expression of human *GUSB* in canine MPS VII models [192]. Intravenous administration of AAV2/8 ARSB (arylsulfatase B) showed a favorable outcome in the liver of an MPS VI feline model, but the cornea was not evaluated [193].

Despite a wealth of knowledge on the methods for the delivery of transgene to the human cornea and ever-increasing information on the genes associated with various corneal dystrophies with substantial therapeutic potential, gene therapy for corneal dystrophies is yet to be explored.

## 7. Tailored Therapeutics Using Gene Therapy

Various approaches can be considered in order to increase precise and targeted expression of a therapeutic gene. The transgene cassette can be optimized to increase AAV transduction efficiency, vector tropism can be improved using capsid engineering, and an appropriate mode of delivery can be used to minimize off-target expression (Figure 7). Further, the capsid and transgene can be genetically modified to avoid the host immune response and the large-scale production of AAV can also be optimized.

### 7.1. Promoter Selection for Targeted Therapeutics

Small and tissue-specific promoters are important tools for preclinical research, for clinical delivery of gene therapies to avoid unwanted transgene expression at off-target areas, and to ensure cell type-specific gene expression. An effective promoter is important to pilot high and clinically relevant levels of therapeutic gene expression. The tissue specific promoter should allow a convenient vector dose for treatment without immune response or cellular toxicity resulting from high virus dosage. In preclinical research, tissue-specific promoters have been used to rescue many animal models of ocular disease. Studies have shown increased efficacy and safety by limiting unwanted off-target effects using tissue specific promoters. Tissue specific promoters may be considered from either the target gene itself or an unrelated gene with the appropriate expression pattern for the therapy.

Over the past few decades most gene therapy studies have used broad expression promoters such as cytomegalovirus (CMV), chicken beta-actin promoter (CAG), and human ubiquitin C promoter (UbiC). Currently, a small number of promoters specific to the cornea are being used, such as keratin K12 promoter and keratocan promoter [194,195]. Previously, promoters of several keratin genes have been used to drive tissue-specific expression of transgenes in animals. There are almost 30 different keratin proteins identified, of which keratin K12 and K3 are expressed in differentiated and stratified corneal epithelium. A study had showed high functional and tissue specific activity with the use of three 5′ truncated fragments of the keratin K12 promoter in human corneal epithelial cell lines [195]. The stroma of the cornea consists of many cell types, most of the cells being keratocytes [196]. Keratocan is a cornea stroma-specific keratan sulfate proteoglycan (KSPG) expressed in adult vertebrate cornea [195]. Li et al. identified the 3.2kb 5′ flanking promoter region of the keratocan gene and showed that it was able to drive β-galactosidase reporter specifically in the stromal layer of the cornea in adult transgenic mice [197]. Similarly, another study showed that keratocan promoter was capable of driving EGFP expression tissue-specifically in the corneal keratocyte [194].

More strongly regulated control of the protein expression level is required to avoid off target toxicity. This demand paved the way for the development of a variety of synthetic promoters in an effort to target expression in particular corneal cell types of interest as well as to provide physiologic expression of the exogenous transgenes. Recently, a group bioinformatically designed human DNA MiniPromoter (Ple253) for tissue-specific expression of transgene in the corneal stroma. Expression using Ple253 (PITX3) was robust and enriched in corneal stroma after neonatal intravenous delivery at P0. Furthermore, there was no expression in the brain, spinal cord, and heart but moderate expression in the liver and the pancreas. Expression at moderate levels in both the ganglion cell layer (GCL) and inner nuclear layer (INL) was observed after intravenous delivery at both P0 and P4 [198]. Additionally, there are several genes that have restricted expression in different regions of the cornea and could serve as a potential candidate promoter to drive cell-specific expression of a gene of interest. These include keratin-3 from the epithelium, decorin from the stroma [181], and ovary-specific protein from the endothelium [199]. Other potential candidate promoters include Aquaporin 5 (AQP5), which expressed in salivary and lacrimal glands and in corneal epithelium, and AQP1, expressed in corneal endothelium [200]. 

Another group demonstrated the possibility of designing cell-specific promoters in silico for in vivo applications. However, this was for retinal pigment epithelium (RPE) specific expression of the transgene [201]. Nevertheless, these findings demonstrate that with the increase in the quality and volume of ‘omics data and with the progressive development of TFRE database and informatic tools, it is now possible to successfully design synthetic promoters that will facilitate the advancement of more targeted and precise gene expression.

### 7.2. Capsid Engineering

One of the promising strategies to improve the efficacy of the AAV vector system for clinical application is capsid engineering. Various capsid engineering approaches include rational design and in-silico bioinformatic approaches [202]. The differences in protein sequence and structure between various wildtype capsids can lead to differences in transduction efficiency and the cell surface receptors utilized for entry. Further, it can also affect the relative biodistribution as well as the affinity for antibodies. Improving transduction efficiency will, in turn, help reduce the vector dose, lower the risk of host immune responses, and reduce the cost of manufacturing. This concept led to the idea of modifying capsid proteins to improve challenges associated with gene delivery. However, most of the studies to date have focused on modifying capsid to improve transduction efficiency in the retinal layers with little work conducted to improve transduction in the cornea.

Alteration of capsid to improve function involves rational design using the current information of the capsids including its crystal structures [203,204,205], cell surface receptors [205,206], immune system activation [207], infectious pathways [108], and antibody binding epitopes [208,209,210]. Experiments have shown an increase in AAV transduction efficiency with the presence of inhibitors of tyrosine kinases [211,212]. Additionally, it was shown that phosphorylation of tyrosine residue led to capsid degradation [213]. Zhong et al. showed that Y444F and Y730F mutations decreased phosphorylation and subsequent ubiquitination of the capsids, leading to significant improvement in transduction in vitro and in vivo. Another study showed improved transduction by mutating three surface tyrosine residues in murine hepatocytes in vivo [214]. Capsid modifications have also been extensively used to improve transduction efficiency in the retina. A study showed efficient transduction of mouse photoreceptors when four surface tyrosine residues and one surface threonine residue were mutated (Y272F, Y444F, Y500F, Y730F, T491V) [215].

Using emerging tools and technologies, continuous efforts are being made to advance capsid engineering. Furthermore, newer strategies to develop novel capsids are being used including next generation sequencing, AI-based machine learning, and bioinformatic capsid prediction tools that will dramatically accelerate progress towards achieving improved gene delivery and targeted therapeutics.

## 8. Challenges and Safety Aspect of Gene Therapy for Corneal Diseases

Despite significant advancement over the last two decades, more research is needed to address the challenges associated with corneal gene therapy. The risk of host immune responses towards the vector remains the largest challenge for AAV-based gene therapies. The capsid protein can elicit an immune response and lead to generation of neutralizing antibodies that could prevent the vector from infecting the patient cells and reduce the effectiveness of the treatment; however, given the immune advantaged location of the cornea, this is less of a risk. Furthermore, highly tissue specific expression of the transgene that will increase efficacy and safety by limiting unwanted off-target effect is required. However, across studies, the expression of transgenes is usually lower in corneal tissues compared to muscle, liver, etc., if normalized for dosage, therefore necessitating careful dose escalation studies. High dosage can add to the existing layer of complexity, where to overcome the barrier of delivering the right amount of AAV to targeted cells, higher dose therapies could cause safety concerns in the long term, particularly in the case of acquired diseases. Therefore, special consideration must be given to deliver vectors with “tunable” or “on/off” control expression cassettes. The last and major challenge associated with corneal gene therapy is the high cost of its research, development, and manufacturing. Further evaluation of the appropriate routes of administration, capsid choice, and vector genome designs are still required to advance corneal gene therapy from the preclinical to clinical setting.

## 9. Conclusions

Gene therapy approaches stand on the front line of advanced biomedical research and have matured considerably in the last decade as a new branch of regenerative medicine. Aside from corneal transplants, very few approaches are available to treat corneal diseases. Thus, gene therapy can be considered as a potential approach in treating various corneal conditions as it is able to correct the underlying pathological mechanism of a disease process with prolonged benefits. Identification of several genes involved in various inherited and acquired corneal conditions have further paved the way for the use of gene therapy approaches as a treatment modality for corneal conditions, including corneal neovascularization, corneal fibrosis, corneal graft rejection, and corneal dystrophies. However, further evaluation of the appropriate routes of administration, capsid choice, and vector genome designs are still needed to advance corneal gene therapy from the preclinical to clinical setting as a more precise and targeted therapeutic approach.

## Figures and Tables

**Figure 1 cells-12-01280-f001:**
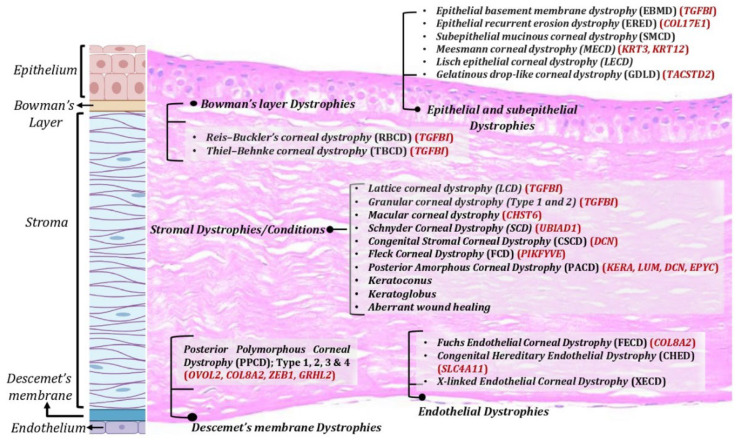
Corneal dystrophies across various corneal layers. Corneal dystrophies affect different layers of the cornea including epithelium, Bowman’s layer, stroma, and Descemet’s membrane and endothelium. Created with Biorender.com (accessed on 14 March 2023).

**Figure 2 cells-12-01280-f002:**
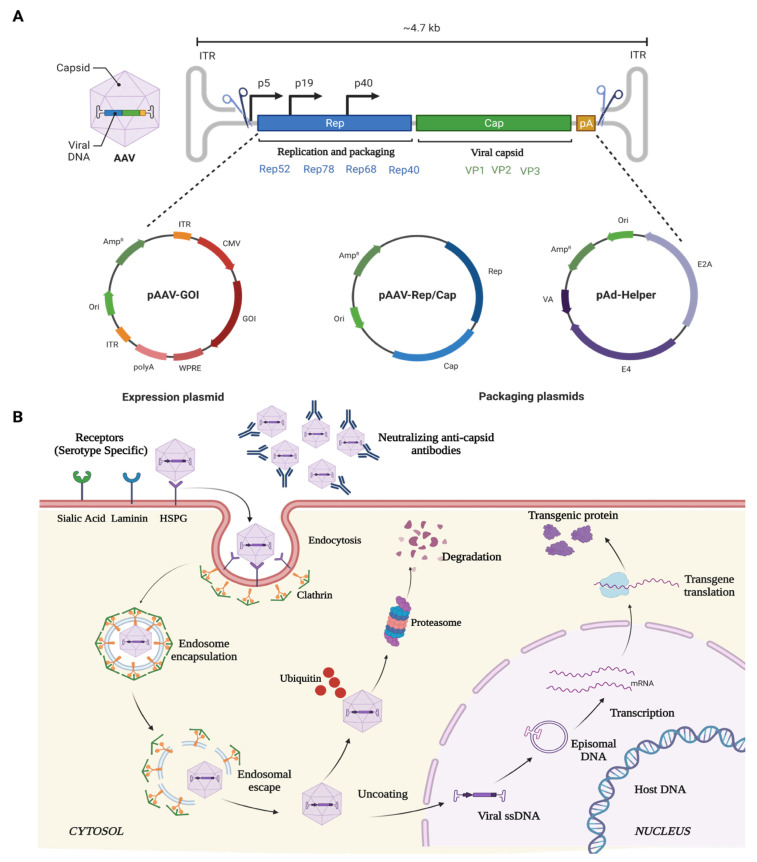
Schematic illustration of (**A**) AAV genome, and (**B**) mechanism of entry, infection, and therapeutic gene production in host cell. Created with Biorender.com (accessed on 14 March 2023).

**Figure 3 cells-12-01280-f003:**
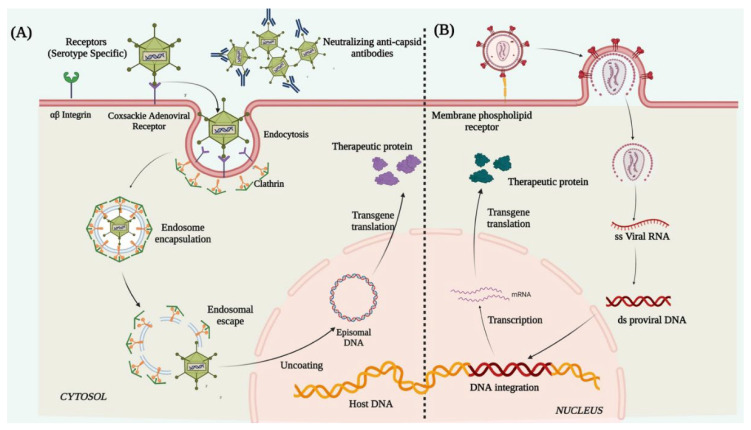
Schematic illustration representing mechanism of action of (**A**) adenovirus and (**B**) lentivirus. Created with Biorender.com (accessed on 14 March 2023).

**Figure 4 cells-12-01280-f004:**
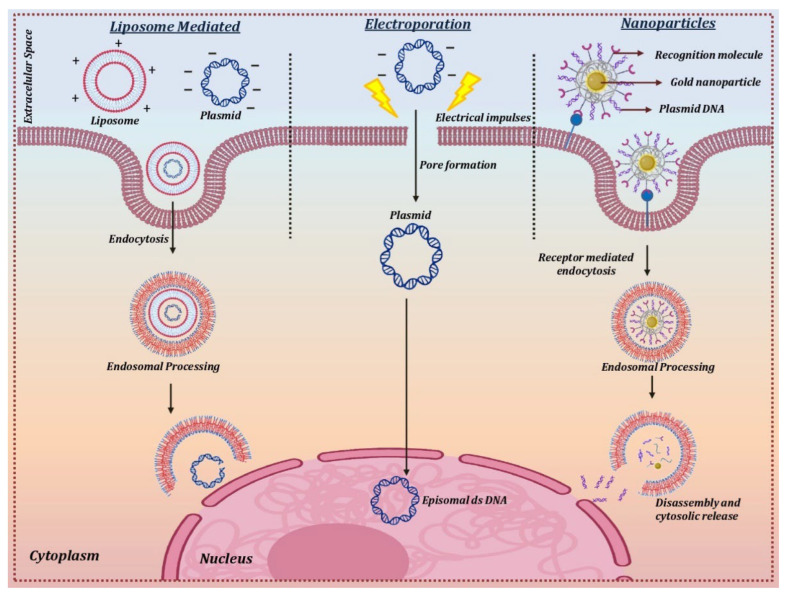
Delivery mechanism for non-viral vectors. Created with Biorender.com (accessed on 14 March 2023).

**Figure 5 cells-12-01280-f005:**
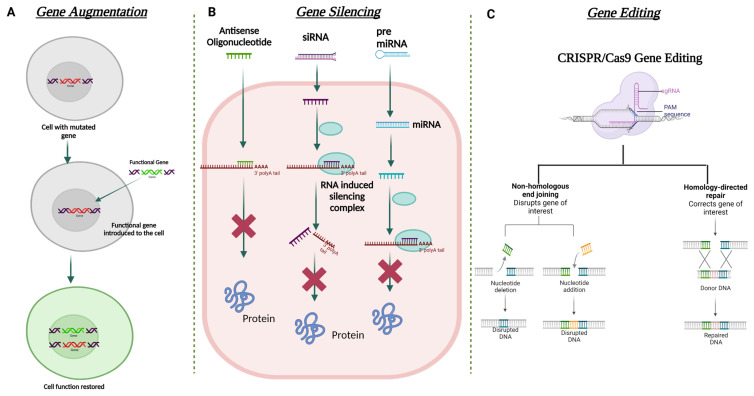
Schematic illustration for the different strategies used in corneal gene therapy. (**A**) Gene augmentation. (**B**) Gene silencing using ASO, siRNA, and miRNA. (**C**) Gene editing using CRISPR Cas9 approach. Created with Biorender.com (accessed on 14 March 2023).

**Figure 6 cells-12-01280-f006:**
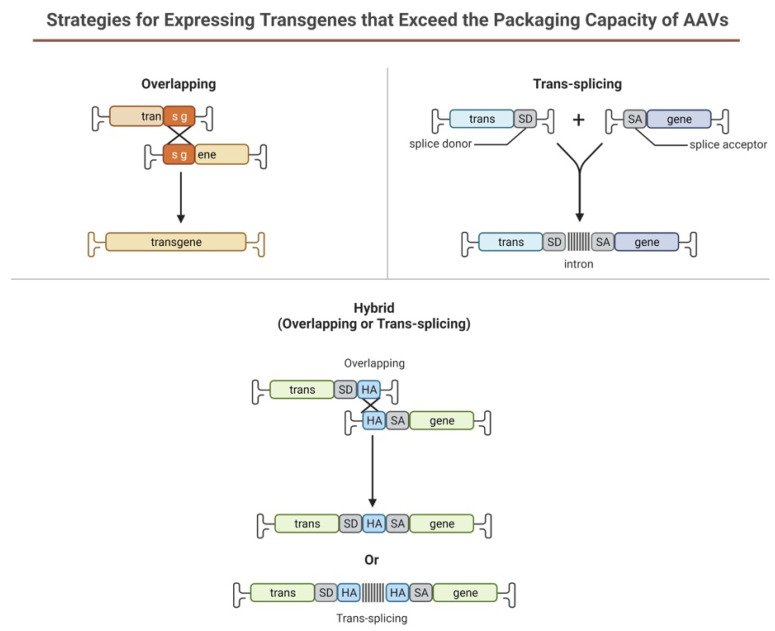
Schematic illustration representing dual AAV approach for large genes. Created with Biorender.com (accessed on 14 March 2023).

**Figure 7 cells-12-01280-f007:**
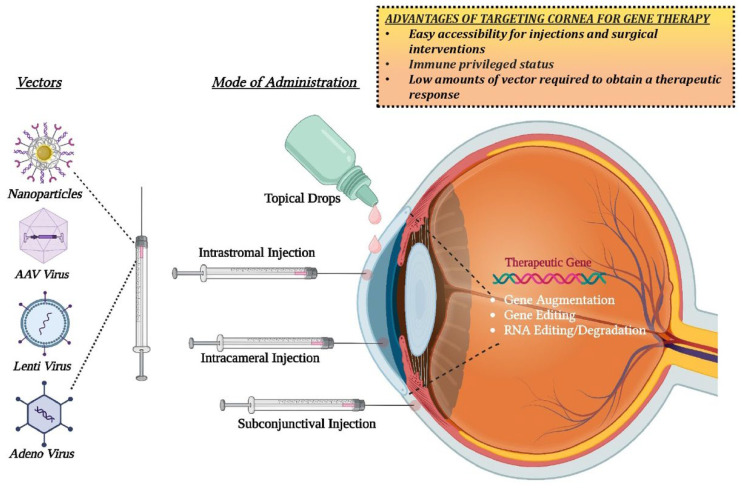
Routes of administration for corneal gene therapy. Created with Biorender.com (accessed on 14 March 2023).

**Table 1 cells-12-01280-t001:** Classification of Corneal dystrophies.

Region	Name of Dystrophy	Gene	Gene Locus	Mode of Inheritance	IC3D Category	Age of Onset	Symptoms	Visual Acuity	Clinical Appearance of the Cornea
**Epithelial and Subepithelial Dystrophies**	Epithelial basement membrane dystrophy (EBMD)	*TGFBI*	5q31	Sporadic	1	Adult	Corneal erosion, slightly distorted vision	Mild visual reduction	Thickening of the epithelium, round or oval opacities, and lines.
	Epithelial recurrent erosion dystrophy (ERED)	*COL17E1*	10q23	Autosomal Dominant	3	1st Decade	Stinging, burning, painful corneal erosion, photophobia	Sometimes impaired	Epithelial erosion
	Subepithelial mucinous corneal dystrophy (SMCD)	*Unknown*	unknown	Autosomal Dominant	4	1st Decade	Painful incidence of recurrent corneal erosion	Progressive loss of vision	Bilateral subepithelial opacities and haze, mostly denser centrally, involving the entire cornea
	Meesmann corneal dystrophy (MECD)	*KRT3, KRT12,*	12q13, 17q12	Autosomal Dominant	1	Early childhood	Mild erosion and reduced sensation of the cornea	Rarely blurred vision	Multiple, tiny, distinct epithelial vesicles extend to the limbus and are mostly cumulated in the interpalpebral area.
	Lisch epithelial corneal dystrophy (LECD)	*Unknown*	Xp22.3	X-linked Dominant	2	Childhood	symptomatic or blurred vision if the pupillary zone is involved	Sometimes impaired	Localized epithelial opacities of various patterns: whorls, bands, flame, feather shaped.
	Gelatinous drop-like corneal dystrophy (GDLD)	*TACSTD2 (M1S1)*	1p32	Autosomal Recessive	1	1st to 2nd Decade	Distorted vision, photophobia, scratchy sensation, redness, tearing	Marked visual impairment	Appearance of subepithelial lesions, indicating extremely hyperpermeable corneal epithelium, Superficial vascularization, stromal opacification
**Bowman Layer Dystrophies**	Reis–Buckler’s corneal dystrophy (RBCD)	*TGFBI*	5q31	Autosomal Dominant	1	Childhood	Painful incidence of recurrent corneal erosion	Progressive deuteriation of vision	Replacement of bowman layer by sheet like connective tissue with granular deposits, which extends to subepithelial stroma.
	Thiel–Behnke corneal dystrophy (TBCD)	*TGFBI*	5q31	Autosomal Dominant	2	Childhood	Painful incidence of recurrent corneal erosion	Gradual visual impairment	Symmetrical subepithelial reticular (honeycomb) opacities in the central cornea; which can progress to deep stromal layers and corneal periphery.
**Stromal Dystrophies**	Lattice corneal dystrophy (LCD)	*TGFBI*	5q31	Autosomal Dominant	1	1st Decade	Stinging, burning, Painful incidence of recurrent corneal erosion	Progressive visual impairment	Thin branching refractile lines or subepithelial ovoid dots in the central cornea, diffuse stromal, ground-glass haze develops later
	Granular corneal dystrophy (Type 1 and 2)	*TGFBI*	5q31	Autosomal Dominant	1	Childhood; early as 2 years of age	Frequent corneal erosion, photophobia, glare.	Decrease in visual acuity as opacification progresses with age.	Well-defined granular opacities are observed that don’t extend to the limbus. Type 2 can add snowflakes and lattice lines between the granules.
	Macular corneal dystrophy (MCD)	*CHST6*	16q22	Autosomal Recessive	1	Childhood	Painful incidence of recurrent corneal erosion, reduced corneal sensitivity, photophobia	Severe visual impairment between 10–30 years.	Thinning of the cornea, in advanced stage corneal endothelium is affected and the Descemet membrane develops guttate excrescences. Limbus to limbus stromal haze, which later spreads to superficial, central, elevated white opacities.
	Schnyder Corneal Dystrophy (SCD)	*UBIAD1*	1p36	Autosomal Dominant	1	Childhood to 2nd or 3rd decade	Reduced corneal sensitivity, glare increases, disproportionate decrease of photopic vision, may have hyperlipoproteinemia (type IIa, III, or IV)	Visual acuity decreases with age	Initial signs include central corneal haze and/or subepithelial crystals (>23 years), arcus lipoids (23–38 years), mid-peripheral panstromal haze (after 38 years)
	Congenital Stromal Corneal Dystrophy (CSCD)	*DCN*	12q21.33	Autosomal Dominant	1	Congenital	Irregular and cloudy appearance of the cornea, reduced visual acuity, increased glare	Moderate to severe visual loss	Diffuse, bilateral, corneal clouding with flake-like, whitish stromal opacities throughout the stroma, pachymetry demonstrates increase in thickness.
	Fleck Corneal Dystrophy (FCD)	*PIKFYVE*	2q34	Autosomal Dominant	1	Congenital	Asymptomatic	Normal	Small, translucent, discoid opacities that are scattered sparsely throughout without affecting the central cornea. Involvement of the asymmetric or unilateral corneal.
	Posterior Amorphous Corneal Dystrophy (PACD)	*KERA, LUM, DCN, EPYC*	12q21.33	Autosomal Dominant	3	1st Decade, early as 16 weeks, possibly congenital nature	Mildly effected visual acuity	Mild visual reduction	Diffused grayish-white sheet like opacities in the posterior part of stroma, corneal thinning (>380 µm), flat corneal topography (<41.00 D) and hyperopia in the centroperipheral form.
**Descemets Membrane and Endothelial Dystrophies**	Fuchs Endothelial Corneal Dystrophy (FECD); early and late-onset	*COL8A2*	1p34.3–p32, 13pTel–13q12.13, 15q, 18q21.2-q21.32	Autosomal Dominant	1, 2	4th decade or later	Epiphora due to recurrent corneal erosion, photophobia, pain, epithelial/stromal edema.	Progressive visual impairment	Diffuse thickening of Descemet membrane with excrescences (guttae). Endothelial cells sparse and atrophic
	Posterior Polymorphous Corneal Dystrophy (PPCD); Type 1, 2, 3 and 4	*OVOL2, COL8A2, ZEB1, GRHL2*	20p11.23, 1p34.3–p32.3, 10p11.2, 8q22.3	Autosomal Dominant	2, 1	Childhood	Stromal clouding, endothelial decomposition that is often asymptomatic	Rarely extensive and progressive visual impairment	Deep corneal lesions that can be nodular, vesicular or blister-like. Edema of the stromal and epithelial layer, endothelial decomposition
	Congenital hereditary endothelial dystrophy	*SLC4A11*	20p13	Autosomal Recessive	1	Congenital	Stromal clouding, blurred vision, photophobia	Blurring vision	Corneal thickening, clouding of the cornea, elevated IOP
	X-linked Endothelial Corneal Dystrophy	*Unknown*	Xq25	X-chromosomal Dominant	2	Congenital	Blurring vision	Blurred vision in males	Cloudy cornea only in males, moon crater–like endothelial changes

*TGFBI*, Transforming growth factor beta-induced; *KRT3*, Keratin 3; *COL8A2*, collagen type VIII alpha 2; *ZEB1*, two-handed zinc-finger homeodomain transcription factor 8; *SLC4A11*, solute carrier family 4 member 11; *PIP5K3,* Phosphatidylinositol-3-phosphate/phosphatidylinositol 5-Kinase type III; *DCN*; Decorin, *UBIAD1*; UbiA prenyltransferase domain containing 1, *CHST6*, Carbohydrate sulfotransferase 6 gene; *GSN*, Gelsolin; *TACSTD2*, tumor associated calcium signal transducer, *KERA*, deletion of keratocan; *LUM*, lumican; *EPYC*, epiphycan; *PIKFYVE*, phosphoinositide kinase, FYVE finger containing; *OVOL2*, Ovo like zinc finger 2; *GRHL2*, Grainyhead-like transcription factor 2.

**Table 2 cells-12-01280-t002:** Advantages and disadvantages of viral and non-viral vectors.

Vectors	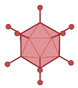 *AdenoVirus*	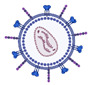 *Lenti Virus*	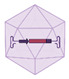 *AAV Virus*	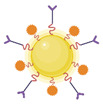 *Non-Viral*
**Advantages**	Non-mutagenicHighly efficient for infecting dividing and non-diving corneal cells in vivo, in vitro and ex vivoHigh titreHigh success rate	Highly efficient for infecting and delivering genes into dividing and non-diving corneal cellsSustained gene expressionBroad tropismWidely useful for in vitro gene therapy concept testing and expression profilingLarger transgene carrying capacity	Highly efficient for infecting and delivering genes into dividing and non-diving corneal cells in vivo, in vitro, and ex vivoSustained and stable transgene expressionLow safety concernMultiple serotypesMinimal/no immune reaction	Highly potent and safe for delivering genes into dividing and non-diving cellsSustained and stable transgene expressionMinimal/ no immune reactionLarge cargo of therapeutic gene transportation capacityAbility to dictate release of therapeutics
**Disadvantages**	Repeat administration of virus is inefficientHigh immune reactionCommon human virus that reduces potencyShort term expressionHigh safety concern	Highly immunogenicRandom integration potential Safety concernHigh titer production requires technical skills	Small insert sizeHigh titer production requires technical skillsConcerns of adeno viral contamination	Short-term transgene expressionPoor efficiencyLow-high immune reaction

**Table 3 cells-12-01280-t003:** Corneal in vivo and ex vivo gene therapy using AAV.

Adeno-Associated Virus Mediated Corneal Gene Therapy										
Vector	Disease	Gene	Promoter	Serotype	Dosage	Model	Species	Mode of Administration	Outcome	Reference
rAAV	Glaucoma	*MMP3*	CMV	AAV-2/9	1 × 10^11^ vgc	In Vivo	WT mouse	Intracameral	Efficient transduction resulted in an increase in aqueous concentration and MMP3 activity, also increased outflow facility and decreased IOP	[118]
rAAV	MPS1-associated Corneal Blindness	IDUA	CMV	AAV8G9	1 × 10^10^ vgc	Ex Vivo	Human	Intrastromal	Efficient widespread transduction resulted in a >10-fold supraphysiological increase in IDUA activity. No significant apoptosis due to AAV or IDUA Overexpression was observed.	[119]
rAAV	MPS1-associated Corneal Blindness	IDUA	CMV	AAV8G9	Increasing volume (50–80 µL); escalating viral dose ranging from (5 × 10^10^–8 × 10^10^) per cornea	In Vivo	MPS I canine model	Intrastromal	Resolution of corneal clouding as early as 1st week, followed by sustained corneal transparency until the end of 25 weeks in eyes of MPS I canines with advanced disease whereas, prevention against the development of advanced corneal changes while restoring clarity was observed in MPS I canines with early corneal disease.	[120]
rAAV	Corneal scarring	*Smad7*	CAG	AAV5	75 μL; (2.67 × 10^13^ μg/mL; *n* = 6)	In Vivo	2–3-month-old New Zealand White female rabbits	Topical drops	Single topical application of AAV vectors in rabbit cornea post-PRK led to a significant decrease in corneal fibrosis and corneal haze.	[113]
rAAV	Corneal scarring	*Decorin*	CAG	AAV5	100 μL; 6.5 × 10^12^ μg/mL	In Vivo	2–3-month-old New Zealand White female rabbits	Topical drops	The stromal haze and fibrosis were decreased in the corneas infected with rAAV. Decorin virions. No immunogenic or toxic response was observed.	[113]
rAAV	corneal inflammation and prevent corneal graft rejection	*HLA-G1*	JET	AAV8G9	2.4 × 10^10^ vgc total dose	In Vivo	Naive Lewis rats	Intrastromal	Corneal intrastromal delivery of AAV.HLA-G subsequently reduced corneal inflammation, vascularization and fibrosis post-corneal injury.	[121]
rAAV	Corneal Neovascularization	*Angiostatin*	CMV	-	5 µL; 1 × 10^10^ viral particles	In Vivo	10–12 weeks old, Male Sprague-Dawley rats	Subconjunctival	Subconjunctival delivery of AAV. Angiostatin showed a significant reduction of alkali burn-induced corneal angiogenesis.	[122]
rAAV	Corneal Neovascularization	*Flt-1*	CMV	AAV9	4 × 10^11^ particles/mL	In Vivo	Rats	Anterior chamber	The subsequent reduction in the development of corneal neovascularization in the stroma of cauterised rats by 36% in comparison to the control group.	[123]
rAAV	Corneal Neovascularization	*miR-204*	CMV	rAAVrh.10	3.6 × 10^10^ GCs (Intrastromal), 3.6 × 10^10^ GCs (Subconjunctival)	In Vivo	6- to 8-week-old female C57BL/6J mice	Intrastromal injection; subconjunctival injection	Attenuation of corneal neovascularization was observed in the alkali-burned cornea	[124]
rAAV	Corneal scarring/ fibrosis	*Id3*	CAG	AAV5	100 μL; 6.5 × 10^12^ μg/mL	In Vivo	2–3-month-old New Zealand White female rabbits	Topical drops	Attenuation of corneal fibrosis and restoration of corneal transparency was observed. No cellular toxicity was reported.	[125]

**Table 4 cells-12-01280-t004:** Corneal in vivo gene therapy using LVs.

Lentivirus Mediated Corneal Gene Therapy									
Vector	Disease	Gene	Promoter	Dosage	Model	Species	Mode of Administration	Outcome	Reference
rLV	Corneal Neovascularization	*Endostatin/Kringle 5*	CMV	50 µL, approximately 10^8^ virus particles/mL	In Vivo	New Zealand White rabbits	Subconjunctival	Corneas transduced with rLV.E-K-5 vector demonstrated inhibition of neovascularization and graft failure, whereas in control animals an early onset and profound neovascularization was observed. 5/6 animals in the control group had graft failure.	[59]
rLV	Corneal Graft Rejection	*Bcl-xL*	CMV	5.5 × 10^6^ IU/mL	In Vivo	BALB/c mice were used as recipients, and C57BL/6 mice (MHC and multiple minor H disparate) or BALB/c (syngeneic) corneas were used as donors	Topically transducing the cultured corneas ex vivo before transplantation.	Delivery of rLV. Bcl-xL to the corneal endothelium of donor corneas significantly improved and promoted allografts’ survival by preventing the endothelium’s apoptosis.	[130]
rLV	Corneal Graft Rejection	*PD-L1*	Ubi-1	Between 3.4 × 10^7^ and 1 × 10^8^ titration units (TU)/mL	In Vivo	Male Lewis (LEW, RT-1) rats served as recipients of male Dark Agouti (DA, RT-1^avl^) grafts	Topically transducing the cultured corneas ex vivo before transplantation.	A subsequent increase in the expression of PD-L1 levels in corneal cells helped prolong allograft survival with minimal proinflammatory cytokine expression.	[131]
rLV	Corneal Fibrosis	*Smad7*	CMV	1 × 10^4^ IFU/μL	In Vivo	Sprague-Dawley rats, approximately 8 weeks of age	Topical drops	Exogenous expression of Smad7 gene expression resulted in reduced activation of the TGFβ/Smad signalling caused by the downregulation of phosphorylation of Smad2. Cell proliferation and fibrotic markers were also inhibited by Smad7.	[132]
rLV	Corneal Fibrosis	*IL-10*	SV40	6.3 × 10^6^ TU/mL	In Vivo	Ovine	Topically transducing the cultured corneas ex vivo before transplantation.	Prolonged survival of corneal allograft by a median of 7 days in recipient corneas transduced with lentivirus containing the therapeutic gene, compared to the control group.	[133]
rLV	Corneal Fibrosis	*p35*	CMV	3 × 10^5^ IU/mL	In Vivo	8 to 12 weeks old male C57BL/6 (B6) and (B/C) mice	Topically transducing the cultured corneas ex vivo before transplantation	Exogenous expression of the p35 gene reduced the graft-mediated immune response due to decreased CD4+ T-cells expression in the cornea.	[134]

**Table 5 cells-12-01280-t005:** Corneal in vivo and ex vivo gene therapy using Ads.

Adenovirus Mediated Corneal Gene Therapy									
Vector	Disease	Gene	Promoter	Dosage	Model	Species	Mode of Administration	Outcome	Reference
Ad	Corneal scarring	*BMP-7*	CAG	3 µL, 2 × 10^7^ PFU/mL	In Vivo	C57BL/6 Mice	Topical	Overexpression of BMP-7 subsequently reduced the scarring of the alkali burn corneas post-20 days of infection.	[138]
Ad	Corneal scarring	*Smad7*	CAG	1 × 10^7^ PFU/μL	In Vivo	C57BL/6 Mice	Topical	Exogenous expression of Smad7 in the burned corneal tissue resulted in reduced activation or blocking of the Smad3 signalling and nuclear factor-κB signalling via RelA/p65.	[139]
Ad	Corneal Neovascularization	*Vasohibin-1*	MMP-1	1 × 10^9^ PFU/μL	In Vivo	6 to 8 weeks old female BALB/c mice	Subconjunctival	Subconjunctival delivery of the therapeutic gene significantly reduced the scarring of the alkali burn corneas	[140]
Ad	Corneal Neovascularization	*Flk-1*	CMV	2 µL, 3 × 10^8^ PFU/μL	In Vivo	6–8 weeks old Sprague–Dawley rats	Anterior chamber	Significant inhibition of neovascularization was observed in the cauterized rat in comparison to the control group.	[141]
Ad	Corneal Fibrosis	*PPAR-γ*	CAG	1 × 10^7^ PFU/μL	In Vivo		Not specified, possibly topical	Upon overexpression of the transgene in alkali-burned mouse cornea, subsequent suppression of the pro-fibrotic factors and promotion of epithelial healing was observed.	[142]
Ad	Corneal wound healing	*c-met*	CMV	1 × 10^8^ PFU/μL	Ex Vivo	Human	Topically transducing the cultured corneas ex vivo	Transduction of diabetic corneas c-met with c-met restored the HGF signalling, normalized the diabetic marker patterns, and accelerated the wound healing process	[143]
Ad	Corneal Graft Rejection		CMV	1 × 10^8^ PFU/μL	In Vivo	Lewis rats served as recipients of female rats of Dark Agouti	Topically transducing the cultured corneas ex vivo before transplantation.	Successful prevention of allogeneic graft rejection in corneal transplantation	[56]
Ad	Corneal Graft Rejection	*IL-10*	CMV	6.6 × 10^2^–6.6 × 10^8^ PFU/μL	In Vivo	Sheep	Topically transducing the cultured corneas ex vivo before transplantation.	Overexpression of the immunomodulatory cytokine IL-10 showed significantly prolonged corneal allograft survival and reduced incidence of graft rejection.	[144]
Ad	Corneal Graft Rejection	*IL-12 p40*	CMV	0.8–1 × 10^7^ PFU/cornea	In Vivo	Sheep	Topically transducing the cultured corneas ex vivo before transplantation.	Local intraocular production of p40 IL-12 increased the corneal graft survival in comparison to the control group which got rejected at the median of 20 days.	[145]

## Data Availability

Data will be available on request.

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
