# Peer review of "Corneal Regeneration Using Gene Therapy Approaches"

_cells, 2023, doi:10.3390/cells12091280_

Round 1
Reviewer 1 Report
The paper is a well structured review of the state of the art about gene therapy for corneal pathologies. The topic is very interesting and represents a promising approach for future management of corneal diseases. The exposition is very clear and exhaustive. The corneal pathologies considered are various with extremely different incidence. To better clarify the clinical relevance of this promising terapeuthical of this promising terapeuthical approach, I suggest to differentiate the various disease examined , giving more evidence to that , as keratoconus, related to an important clinical and social impact.
Author Response
Thank you for giving us the opportunity to submit a revised draft of the manuscript “Corneal Regeneration using Gene Therapy”. We appreciate the time and effort taken to provide valuable feedback on our manuscript. We are grateful to the reviewers for their insightful comments and constructive suggestions on our paper. We have noted each comment and have incorporated changes to reflect most of the suggestions provided by the reviewers. All revisions to the manuscript have been marked up using the “Track Changes” function. Here is a point-by-point response to the reviewers' comments and concerns. The addressed comments have been highlighted in blue.
Reviewer 1:
The paper is a well structured review of the state of the art about gene therapy for corneal pathologies. The topic is very interesting and represents a promising approach for future management of corneal diseases. The exposition is very clear and exhaustive. The corneal pathologies considered are various with extremely different incidence. To better clarify the clinical relevance of this promising terapeuthical of this promising terapeuthical approach, I suggest to differentiate the various disease examined , giving more evidence to that , as keratoconus, related to an important clinical and social impact.
Author Response: Thank you for your valuable comments, appreciation and suggestions. To improve the readability of the manuscript, we have further categorized the various corneal diseases / conditions that can be potentially treated using gene therapy (under the section- 2).
As suggested by the reviewer, we have included the disease burden of corneal conditions and the socio-economic impact of keratoconus on the patients and their families. We have also emphasised on the contextual importance and need for alternate therapies (under section 2 and 3). We have also added more information throughout the manuscript.
Additional clarifications:
In addition to the above comments, all the tables have been added in the tables format to the main text after its first citation. For the figures that were made on BioRender, this information is added in the figure caption. All the figures are original and therefore do not require access data.

Reviewer 2 Report
This manuscript aim to discuss the multiple aspects of corneal gene therapy. Unfortunately, the whole manuscript should be carefully read and shortened, as there are many unnecessary information making text very hard to follow. I suggest to focus only on the gene therapy not a pathophysiology of diseases, „classic“ treatment options etc.
Keratoconus should be omitted as it is not a Mendelian disorder
The list of corneal dystrophies and Figure 1 must be updated. I recommend the last version of IC3D paper, for example OVOL2 (discovered as cause ofPPCD1 in 2016) and GRHL2 (PPCD4, 2018) are missing, and also CHED1 does not exist since 2016.
All genes names should be in italics.
Tables are very blurry.
Author Response
Thank you for giving us the opportunity to submit a revised draft of the manuscript “Corneal Regeneration using Gene Therapy”. We appreciate the time and effort taken to provide valuable feedback on our manuscript. We are grateful to the reviewers for their insightful comments and constructive suggestions on our paper. We have noted each comment and have incorporated changes to reflect most of the suggestions provided by the reviewers. All revisions to the manuscript have been marked up using the “Track Changes” function. Here is a point-by-point response to the reviewers' comments and concerns. The addressed comments have been highlighted in blue.
Reviewer 2:
Comment 1: This manuscript aim to discuss the multiple aspects of corneal gene therapy. Unfortunatemnly, the whole manuscript should be carefully read and shortened, as there are many unnecessary information making text very hard to follow. I suggest to focus only on the gene therapy not a pathophysiology of diseases, „classic“ treatment options etc.
Author Response: We understand the reviewer’s concern and have shortened the sections that focus on pathophysiology of disease and classic treatment options (Section 3 and 4). We have also added sub-headings to structure the manuscript for better readability. We have made several deletions and additions throughout the manuscript across various sections. However, we have retained the sections rather than removing them completely since they help set up the background information for the need for advanced therapies such as gene therapy. However, we have strengthened further, the gene therapy aspects with the addition of a new table enumerating the pros and cons of various gene therapy vectors (Table 2).
Comment 2: Keratoconus should be omitted as it is not a Mendelian disorder
Author Response: The reviewer is correct in pointing out that Keratoconus is not a Mendelian disorder. However, gene therapy applications are not limited to Mendelian diseases, case in example are the various gene therapy trials currently running across diseases such as cancer, metabolic diseases and eye diseases such as AMD. In the literature we have cited, corneal gene therapy has been done in various models for acquired corneal conditions such as wound healing, neovascularisation, etc. Thus, there is much potential for using gene therapy in a gene agnostic, but disease specific manner in multifactorial diseases. We have now clarified multifactorial diseases separately now and included keratoconus under that since it is a most common disease with an unmet clinical need for alternative therapy (Section 2.2). Further, since genetic factors are thought to be involved in Keratoconus and several studies have identified gene mutations responsible for Keratoconus, it makes gene therapy a potential treatment option for Keratoconus. Hence, we believe that including Keratoconus here would be appropriate and within the realm of the review topic as well as the special issue.
Comment 3: The list of corneal dystrophies and Figure 1 must be updated. I recommend the last version of IC3D paper, for example OVOL2 (discovered as cause ofPPCD1 in 2016) and GRHL2 (PPCD4, 2018) are missing, and also CHED1 does not exist since 2016.
Author Response: Thank you for pointing this out. In the revised version of the manuscript, we have updated Table 1 and Figure 1 based on the latest version of IC3D paper. We have also included the genes mentioned.
Comment 4: All genes names should be in italics.
Author Response: Thank you for pointing this out, we apologise for the oversight. In the revised version of the manuscript, we have italicized all the gene names.
Comment 5: Tables are very blurry.
Author Response: Thank you for pointing this out. In the revised version of the manuscript, we have made sure the tables are clear and as per the journal format.
Additional clarifications:
In addition to the above comments, all the tables have been added in the tables format to the main text after its first citation. For the figures that were made on BioRender, this information is added in the figure caption. All the figures are original and therefore do not require access data.

Round 2
Reviewer 2 Report
Authors submitted a revised version of manuscript. However, the manuscript did not improved markebly. It is stil very hard to follow. I recommend authors to focus only on monogenic diseases.
From the response letter: "Further, since genetic factors are thought to be involved in Keratoconus and several studies have identified gene mutations responsible for Keratoconus, it makes gene therapy a potential treatment option for Keratoconus." This is false, there are only GWAS results, that showed association of SNPs and keratoconus. It means that these are risk or protection factors, not a cause of disease.
Table 1 is badly formatted.
Tables should contain reference, not the PMID reference number.
